# Bypassing the Kohn-Sham equations with machine learning

Felix Brockherde[1,2], Leslie Vogt [3], Li Li [4], Mark E. Tuckerman[3,5,6], Kieron Burke[4,7] & Klaus-Robert Müller[1,8,9]

Last year, at least 30,000 scientific papers used the Kohn–Sham scheme of density functional theory to solve electronic structure problems in a wide variety of scientific fields. Machine learning holds the promise of learning the energy functional via examples, bypassing the need to solve the Kohn–Sham equations. This should yield substantial savings in computer time, allowing larger systems and/or longer time-scales to be tackled, but attempts to machine-learn this functional have been limited by the need to find its derivative. The present work overcomes this difficulty by directly learning the density-potential and energy-density maps for test systems and various molecules. We perform the first molecular dynamics simulation with a machine-learned density functional on malonaldehyde and are able to capture the intramolecular proton transfer process. Learning density models now allows the construction of accurate density functionals for realistic molecular systems.

[1] Machine Learning Group, Technische Universität Berlin, Marchstraße 23, 10587 Berlin, Germany. [2] Max-Planck-Institut für Mikrostrukturphysik, Weinberg 2, 06120 Halle, Germany. [3] Department of Chemistry, New York University, New York, NY 10003, USA. [4] Department of Physics and Astronomy, University of California, Irvine, CA 92697, USA. [5] Courant Institute of Mathematical Science, New York University, New York, NY 10003, USA. [6] NYU-ECNU Center for Computational Chemistry at NYU Shanghai, 3663 Zhongshan Road North, Shanghai 200062, China. [7] Department of Chemistry, University of California, Irvine, CA 92697, USA. [8] Department of Brain and Cognitive Engineering, Korea University, Anam-dong, Seongbuk-gu, Seoul 136-713, Republic of Korea. [9] Max-Planck-Institut für Informatik, Stuhlsatzenhausweg, 66123 Saarbrücken, Germany. Correspondence and requests for materials should be addressed to M.E.T. (email: mark.tuckerman@nyu.edu) or to K.B. (email: kieron@uci.edu) or to K.-R.M. (email: klaus-robert.mueller@tu-berlin.de)

**K**ohn–Sham density functional theory[1] (KS-DFT) is now enormously popular as an electronic structure method in a wide variety of fields[2]. Useful accuracy is achieved with standard exchange–correlation (XC) approximations, such as generalized gradient approximations[3] and hybrids[4]. Such calculations are playing a key role in the materials genome initiative[5], at least for weakly correlated materials[6].

There has also been a recent spike of interest in applying machine learning (ML) methods in the physical sciences[7–11]. The majority of these applications involve predicting properties of molecules or materials from large databases of KS-DFT calculations[12–15]. A few applications involve finding potential energy surfaces within molecular dynamics (MD) simulations[16–19]. Fewer still have focussed on finding the functionals of DFT as a method of performing KS electronic structure calculations without solving the KS equations[20–23,24]. If such attempts could be made practical, the possible speed-up in repeated DFT calculations of similar species, such as occur in ab initio MD simulations, is enormous.

A key difficulty has been the need to extract the functional derivative of the non-interacting kinetic energy. The non-interacting kinetic energy functional $T_s[n]$ of the electron density $n$ is used in two distinct ways in a KS calculation[1], as illustrated in Fig. 1: (i) its functional derivative is used in the Euler equation which is solved in the self-consistent cycle and (ii) when self-consistency is reached, the ground-state energy of the system is calculated by $E[n]$, an orbital-free (OF) mapping. The solution of the KS equations performs both tasks exactly. Early results on simple model systems showed that ML could provide highly accurate values for $T_s[n]$ with only modest amounts of training[20], but that the corresponding functional derivatives are too noisy to yield sufficiently accurate results to (i). Subsequent schemes overcome this difficulty in various ways, but they typically lose a factor of 10 or more in accuracy[22], and their computational cost can increase dramatically with system complexity.

Here we present an alternative ML approach, in which we replaced the Euler equation by directly learning the Hohenberg–Kohn (HK) map $v(\mathbf{r}) \rightarrow n(\mathbf{r})$ (*red line* in Fig. 1a) from the one-body potential of the system of interest to the interacting ground-state density, i.e., we establish an ML-HK map. We show that this map can be learned at a much more modest cost than either previous ML approaches to find the functional and its derivative (ML-OF) or direct attempts to model the energy as a functional of $v(\mathbf{r})$ (ML-KS). Furthermore, we show that it can immediately be applied to molecular calculations by calculating the energies of small molecules over a range of conformers. Moreover, as we have already implemented this approach with a standard quantum chemical code (Quantum

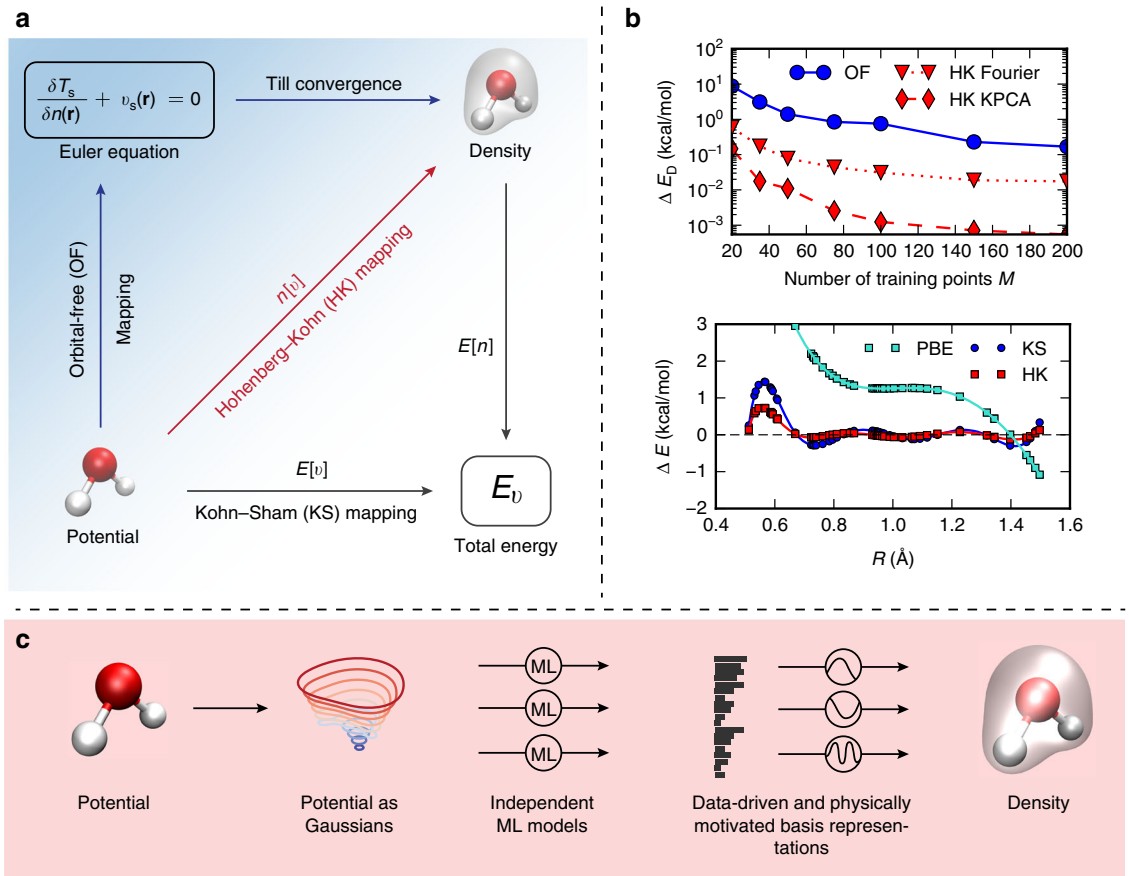

**Fig. 1** Overview of approach and motivation. **a** Mappings used in this paper. The *bottom arrow* represents $E[v]$, a conventional electronic structure calculation, i.e., KS-DFT. The ground-state energy is found by solving KS equations given the external potential, $v$. $E[n]$ is the total energy density functional. The *red arrow* is the HK map $n[v]$ from external potential to its ground state density. **b** (*Top*) How the energy error depends on $M$, the number of training points, for ML-OF and ML-HK with different basis sets for the 1D problem. **b** (*Bottom*) Errors in the PBE energies (relative to exact values) and the ML maps (relative to PBE) as a function of interatomic spacing, $R$, for $H_2$ with $M = 7$. **c** How our Machine Learning Hohenberg–Kohn (ML-HK) map makes predictions. The molecular geometry is represented by Gaussians; many independent Kernel ridge regression models predict each basis coefficient of the density. We analyze the performance of data-driven (ML) and common physical basis representations for the electron density

**Table 1 Energy errors for the 1D data set**

| M | ML-OF | | | | | | ML-HK (grid) | | | | | | ML-HK (other) | | | |
|---|---|---|---|---|---|---|---|---|---|---|---|---|---|---|---|---|
| | $\Delta E$ | | $\Delta E_F$ | | $\Delta E_D$ | | $\Delta E$ | | $\Delta E_D$ | | $\Delta E_D^{ML}$ | | $\Delta E_D$ (Fourier) | | $\Delta E_D$ (KPCA) | |
| | MAE | Max | MAE | Max | MAE | Max | MAE | Max | MAE | Max | MAE | Max | MAE | Max | MAE | Max |
| 20 | 7.7 | 47 | 7.7 | 60 | 8.8 | 87 | 3.5 | 27 | 0.76 | 8.9 | 9.7 | 70 | 0.58 | 8 | 0.15 | 2.9 |
| 50 | 1.6 | 30 | 1.3 | 7.3 | 1.4 | 31 | 1.2 | 7.1 | 0.079 | 0.92 | 0.27 | 2.4 | 0.078 | 0.91 | 0.011 | 0.17 |
| 100 | 0.74 | 17 | 0.2 | 2.6 | 0.75 | 17 | 0.19 | 2.1 | 0.027 | 0.43 | 0.18 | 2.4 | 0.031 | 0.42 | 0.0012 | 0.028 |
| 200 | 0.17 | 2.9 | 0.039 | 0.6 | 0.17 | 2.9 | 0.042 | 0.59 | 0.0065 | 0.15 | 0.02 | 0.46 | 0.017 | 0.14 | 0.00055 | 0.015 |

Errors are given in kcal/mol for various M, the number of training points. Displayed are the energy error, $\Delta E$, the functional-driven error, $\Delta E_F$, the density-driven error, $\Delta E_D$, and its approximation, $\Delta E_D^{ML}$

Espresso[25]) using a standard DFT approximation (Perdew–Burke–Ernzerhof (PBE)), this can now be tried on much larger scales.

The ML-HK map reflects the underlying computational approach used to generate a particular electron density, but is not restricted to any given electronic structure method. Many molecular properties, including but not limited to the energy, are dependent on the electron density, making the ML-HK map more versatile than a direct ML-KS mapping. We also establish that densities can be learned with sufficient accuracy to distinguish between different DFT functionals, thus providing a route to future functional development via generation of precise densities for a range of molecules and conformations.

## Results

We will first outline theoretical results, most prominently the ML-HK map, and then illustrate the approach with simulations of one-dimensional (1D) systems and three-dimensional (3D) molecules.

**ML-HK map**. Previous results show that for an ML-OF approach, the accuracy of ML-KS kinetic energy models $T_s^{ML}[n]$ improve rapidly with the amount of data. However, minimizing the total energy via gradient descent requires the calculation of the gradient of the kinetic energy model $T_s^{ML}$ (Fig. 1) and calculating this gradient is challenging. Due to the data-driven nature of, e.g., kernel models, the machine-learned kinetic energy functional has no information in directions that point outside the data manifold[26]. This heavily influences the gradient to an extent that it becomes unusable without further processing[20]. There have been several suggestions to remedy this problem, but all of them share a significant loss in accuracy compared to $T_s[n]$[21, 22, 27].

Here, we propose an interesting alternative to gradients and the ML-OF approach. Recently, it has been shown that the HK map for the density as a functional of the potential can be approximated extremely accurately using semiclassical expressions[28]. Such expressions do not require the solution of any differential equation, and become more accurate as the number of particles increases. Errors can be negligible even for just two distinct occupied orbitals.

Inspired by this success, we suggest how one might circumvent the kinetic energy gradient and directly train a multivariate ML model. We name this the ML-HK map:

$$n^{ML}[v](x) = \sum_{i=1}^{M} \beta_i(x) k(v, v_i). \tag{1}$$

Here each density grid point is associated with a group of model weights $\beta$. Training requires solving an optimization problem for each density grid point. Although this is possible in 1D, it rapidly

becomes intractable in 3D, as the number of grid points grows cubically.

The use of a basis representation for the densities, as in

$$n^{ML}[v](x) = \sum_{l=1}^{L} u^{(l)}[v] \phi_l(x), \tag{2}$$

renders the problem tractable even for 3D. A ML model that predicts the basis function coefficients $u^{(l)}[v]$ instead of the grid points is then formulated.

Predicting the basis function coefficients not only makes the ML model efficient and allows the extension of the approach to 3D but also permits regularization, e.g., to smooth the predicted densities by removing the high-frequency basis functions, or to further regularize the ML model complexity for specific basis functions.

For orthogonal basis functions, the ML model reduces to several independent regression models and admits an analytical solution analogous to kernel ridge regression (KRR, see Supplementary Eq. (5)):

$$\boldsymbol{\beta}^{(l)} = \left( \mathbf{K}_{\boldsymbol{\sigma}^{(l)}} + \lambda^{(l)} \mathbf{I} \right)^{-1} \mathbf{u}^{(l)}, \quad l = 1, \dots, L. \tag{3}$$

Here, for each basis function coefficient, $\lambda^{(l)}$ are regularization parameters and $\mathbf{K}_{\boldsymbol{\sigma}^{(l)}}$ is a Gaussian kernel with kernel width $\boldsymbol{\sigma}^{(l)}$. The parameters $\lambda^{(l)}$ and $\sigma^{(l)}$ can be chosen individually for each basis function via independent cross-validation (see refs. [12, 29]). This ML-HK model avoids prior gradient descent procedures and, with it, the necessity to "de-noise" the gradients. Due to the independence of Eq. (3) for each $l$, the solution scales favourably.

**Functional and density-driven error**. How can the performance of the ML-HK map be measured? It has recently been shown how to separate out the effect of the error in the functional $F$ and the error in the density $n(\mathbf{r})$ on the resulting error in the total energy of any approximate, self-consistent DFT calculation[30]. Let $\tilde{F}$ be an approximation of the many body functional $F$, and $\tilde{n}(\mathbf{r})$ the approximate ground-state density when $\tilde{F}$ is used in the Euler equation. Defining $\tilde{E}[n] = \tilde{F}[n] + \int d^3r \, n(\mathbf{r}) \, v(\mathbf{r})$ yields

$$\Delta E = \tilde{E}[\tilde{n}] - E[n] = \Delta E_F + \Delta E_D \tag{4}$$

where $\Delta E_F = \tilde{F}[n] - F[n]$ is the functional-driven error, while $\Delta E_D = \tilde{E}[\tilde{n}] - \tilde{E}[n]$ is the density-driven error. In most DFT calculations, $\Delta E$ is dominated by $\Delta E_F$. The standard DFT approximations can, in some specific cases, produce abnormally large density errors that dominate the total error. In such situations, using a more accurate density can greatly improve the result[30–32]. We will use these definitions to measure the accuracy of the ML-HK map.

**1D potentials**. The following results demonstrate how much more accurate ML is when applied directly to the HK map. The box problem originally introduced in Snyder et al.[20] is used to illustrate the principle. Random potentials consisting of three Gaussian dips were generated inside a hard-wall box of length 1 (atomic units), and the Schrödinger equation for one electron was solved extremely precisely. Up to 200 cases were used to train an ML model $T_s^{ML}[n]$ for the non-interacting kinetic energy functional $T_s[n]$ via KRR (for details, see Supplementary Methods).

To measure the accuracy of an approximate HK map, the analysis of the previous section is applied to the KS DFT problem. Here $F$ is just $T_s$, the non-interacting kinetic energy, and

$$\Delta E_F = \tilde{T}_s[n] - T_s[n], \tag{5}$$

i.e., the error made in an approximate functional on the exact density. Table 1 on the left gives the errors made by ML-OF for the total energy, and its different components, when the density is found from the functional derivative. This method works by following a gradient descent of the total energy functional based on the gradient of the ML model $T_s^{ML}$,

$$n^{(j+1)} = n^{(j)} - \epsilon P\left(n^{(j)}\right) \frac{\delta}{\delta n} E^{ML}\left(n^{(j)}\right), \tag{6}$$

where $\epsilon$ is a small number and $P(n^{(j)})$ is a localized PCA projection to de-noise the gradient. Here, and for all further 1D results, we use

$$E^{ML}[n] = T_s^{ML}[n] + \int dx\, n(x)\, v(x). \tag{7}$$

The density-driven contribution to the error, $\Delta E_D$, which we calculate exactly here using the von Weizsäcker kinetic energy[33], is always comparable to, or greater than, the functional-driven error, $\Delta E_F$, due to the poor quality of the ML functional derivative[20]. The calculation is abnormal and can be greatly improved by using a more accurate density from a finer grid. As the number of training points $M$ grows, the error becomes completely dominated by the error in the density. This shows that the largest source of error lies in the use the ML approximation of $T_s$ to find the density by solving the Euler equation.

The next set of columns analyzes the ML-HK approach, using a grid basis. The left-most of these columns shows the energy error we obtain by utilizing the ML-HK map:

$$\Delta E = \left| E^{ML}\left[n^{ML}[v]\right] - E \right|. \tag{8}$$

Note that both ML models, $T_s^{ML}$ and $n^{ML}$, have been trained using the same set of $M$ training points.

The ML-HK approach is always more accurate than ML-OF, and its relative performance improves as $M$ increases. The next column reports the density-driven error, $\Delta E_D$, which is an order-of-magnitude smaller than that of ML-OF. Lastly, we list an estimate to the density-driven error

$$\Delta E_D^{ML} = \left| E^{ML}\left[n^{ML}[v]\right] - E^{ML}[n] \right|, \tag{9}$$

which uses the ML model $T_s^{ML}$ for the kinetic energy functional in 1D. This proxy is generally a considerable overestimate (a factor of 3 too large), so that the true $\Delta E_D$ is always significantly smaller. We use it in subsequent calculations (where we cannot calculate $T_s^{ML}$) to (over-)estimate the energy error due to the ML-HK map.

The last set of columns are density-driven errors for other basis sets. Three variants of the ML-HK map were tested. First, direct prediction of the grid coefficients: in this case, $u_i^{(l)} = n_i(x_l)$, $l = 1, ..., G$. Five hundred grid points were used, as in Snyder, et al.[20]. This variant is tested in 1D only; in 3D the high

dimensionality is prohibitive. Second, a common Fourier basis is tested. The density can be transformed efficiently via the discrete Fourier transform, using 200 Fourier basis functions in total. In 3D, these basis functions correspond to plane waves. The back-projection $u \mapsto n$ to input space is simple, but although the basis functions are physically motivated, they are very general and not specifically tailored to density functions. The performance is almost identical to the grid, on average, although maximum errors are much less. For $M = 20$, the error that originates from the basis representation starts to dominate. This is a motivation for exploring, third, a Kernel PCA (KPCA) basis[34]. KPCA[35] is a popular generalization of PCA that yields basis functions that maximize variance in a higher dimensional feature space. The KPCA basis functions are data-driven, and computing them requires an eigen-decomposition of the Kernel matrix. Good results are achieved with only 25 KPCA basis functions. The KPCA approach gives better results because it can take the non-linear structure in the density space into account. However, it introduces the pre-image problem: it is not trivial to project the densities from KPCA space back to their original (grid) space (Supplementary Note 1). It is thus not immediately applicable to 3D applications.

**Molecules**. We next apply the ML-HK approach to predict electron densities and energies for a series of small molecules. We test the ML models on KS-DFT results obtained using the PBE XC functional[36] and atomic pseudopotentials with the projector augmented-wave (PAW) method[37, 38] in the Quantum ESPRESSO code[25]. As the ML-OF approach applied in the previous section becomes prohibitively expensive in 3D due to the poor convergence of the gradient descent procedure (Supplementary Note 2), we compare the ML-HK map to the ML-KS approach. This approach models the energy directly as a functional of $v(\mathbf{r})$, i.e., it trains a model

$$E^{ML}[v] = \sum_{i=1}^{M} \alpha_i k(v_i, v) \tag{10}$$

via KRR (for details, see Supplementary Methods).

We also apply the ML-HK map with Fourier basis functions. Instead of a $T_s^{ML}[n]$ model, we learn an $E^{ML}[n]$ model

$$E^{ML}[n] = \sum_{i=1}^{M} \alpha_i k(n_i, n), \tag{11}$$

which avoids implementing the potential energy and XC functionals.

Both approaches require the characterization of the Hamiltonian by its external potential. The external (Coulomb) potential diverges for 3D molecules and is, therefore, not a good feature to measure the distance in ML. Instead, we use an artificial Gaussian potential of the form

$$v(\mathbf{r}) = \sum_{\alpha=1}^{N^a} \mathbf{Z}_\alpha \exp\left(\frac{-\|\mathbf{r} - \mathbf{R}_\alpha\|^2}{2\gamma^2}\right), \tag{12}$$

where $\mathbf{R}_\alpha$ are the positions, and $\mathbf{Z}_\alpha$ are the nuclear charges of the $N^a$ atoms. The Gaussian potential is used for the ML representation only. The width $\gamma$ is a hyper-parameter of the algorithm. The choice is arbitrary but can be cross-validated. We find reasonable results with $\gamma = 0.2$ Å. The idea of using Gaussians to represent the external potential has been used previously[39]. The Gaussian potential is discretized on a coarse grid with grid spacing $\Delta = 0.08$. To use the discretized potential in the Gaussian kernel, we flatten it into a vector form and thus use a tensor Frobenius norm.

**Table 2 Prediction errors on H₂ and H₂O**

| Molecule | $M$ | ML-KS | | | | ML-HK | | | | | |
|---|---|---|---|---|---|---|---|---|---|---|---|
| | | $\Delta E$ | | $\Delta R_o$ | $\Delta\theta_o$ | $\Delta E$ | | $\Delta E_D^{ML}$ | | $\Delta R_o$ | $\Delta\theta_o$ |
| | | MAE | Max | | | MAE | Max | MAE | Max | | |
| H₂ | 5 | 1.3 | 4.3 | 2.2 | — | 0.70 | 2.9 | 0.18 | 0.54 | 1.1 | — |
| | 7 | 0.37 | 1.4 | 0.23 | — | 0.17 | 0.73 | 0.054 | 0.16 | 0.19 | — |
| | 10 | 0.080 | 0.41 | 0.23 | — | 0.019 | 0.11 | 0.017 | 0.086 | 0.073 | — |
| H₂O | 5 | 1.4 | 5.0 | 2.1 | 2.2 | 1.1 | 4.9 | 0.056 | 0.17 | 2.3 | 3.8 |
| | 10 | 0.27 | 0.93 | 0.63 | 1.9 | 0.12 | 0.39 | 0.099 | 0.59 | 0.12 | 0.38 |
| | 15 | 0.12 | 0.47 | 0.19 | 0.41 | 0.043 | 0.25 | 0.029 | 0.14 | 0.064 | 0.23 |
| | 20 | 0.015 | 0.064 | 0.043 | 0.16 | 0.0091 | 0.060 | 0.011 | 0.058 | 0.024 | 0.066 |

Errors are shown for increasing numbers of training points $M$ for the ML-KS and ML-HK approaches. In addition, the estimated density-driven contribution to the error for the ML-HK approach (Eq. (9)) is given. Energies are given in kcal/mol, bond-lengths in pm, and angles in degrees

Our first molecular prototype is H₂, with the only degree of freedom, $R$, denoting the distance between the H atoms. A data set of 150 geometries is created by varying $R$ between 0.5 and 1.5 Å (sampled uniformly). A randomly chosen subset of 50 geometries is designated as the test set and is unseen by the ML algorithms. These geometries are used to measure the out-of-sample error reported below.

The remaining 100 geometries make up the grand training set. To evaluate the performance of the ML-KS map and the ML-HK map, subsets of varying sizes $M$ are chosen out of the grand training set to train the $E^{ML}[v]$ and $n^{ML}[v]$ models, respectively. Because the required training subsets are so small, careful selection of a subset that covers the complete range of $R$ is necessary. This is accomplished by selecting the $M$ training points out of the grand training set so that the $R$ values are nearly equally spaced (see Supplementary Note 3 for details).

For practical applications, it is not necessary to run DFT calculations for the complete grand training set, only for the $M$ selected training points. Strategies for sampling the conformer space and selecting the grand training set for molecules with more degrees of freedom are explained for H₂O and MD simulations later on.

The performance of the ML-KS map and ML-HK map is compared by evaluating $E^{ML}[v]$ that maps from the Gaussian potential to the total energy and the combination of $n^{ML}[v]$ that maps from Gaussian potential to the ground-state density in a 3D Fourier basis representation ($l = 25$) and $E^{ML}[n]$ that maps from density to total energy. The prediction errors are listed in Table 2.

The mean average error (MAE) of the energy evaluated using the ML-HK map is significantly smaller than that of the ML-KS map. This indicates that even for a 3D system, learning the potential-density relationship via the HK map is much easier than directly learning the potential-energy relationship via the KS map.

Figure 1b shows the errors made by the ML-KS and the ML-HK maps. The error of the ML-HK map is smoother than the ML-KS error and is much smaller, even for the most problematic region when $R$ is smaller than the equilibrium bond distance of $R_0 = 0.74$ Å. The MAE that is introduced by the PBE approximation on the H₂ data set is 2.3 kcal/mol (compared to exact CI calculations), i.e., well above the errors of the ML model and verifies that the error introduced by the ML-HK map is negligible for a DFT calculation.

The next molecular example is H₂O, parametrized with three degrees of freedom: two bond lengths and a bond angle. In order to create a conformer data set, the optimized structure ($R_0 = 0.97$ Å, $\theta_0 = 104.2$ using PBE) is taken as a starting point. A total of 350 geometries are then generated by changing each bond length by a uniformly sampled value between ±0.075 Å and varying the

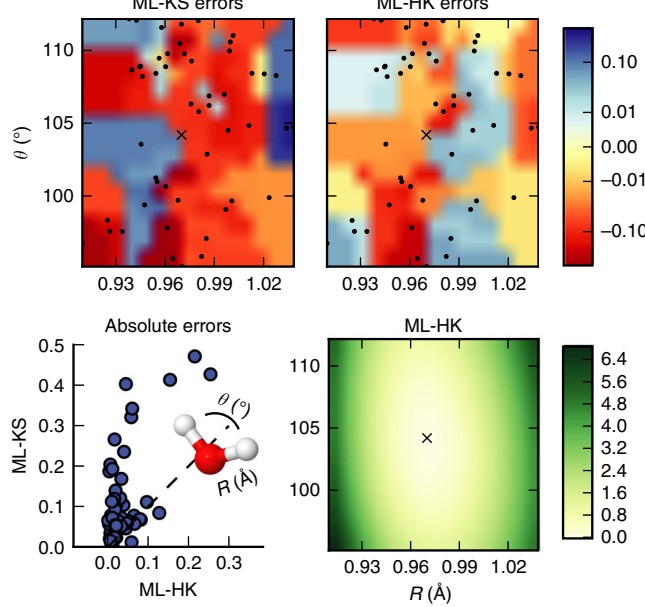

**Fig. 2** Result of the comparison for H₂O. (*Top*) Distribution of energy errors against PBE on the H₂O data set for ML-KS and ML-HK. The errors are plotted on a symmetric *log scale* with a linear threshold of 0.01, using nearest neighbor interpolation from a grid scan for coloring. *Black dots* mark the test set geometries with averaged bond lengths. (*Bottom left*) Comparison of the PBE errors made by ML-HK and ML-KS on the test set geometries. (*Bottom right*) Energy landscape of the ML-HK map for symmetric geometries ($R$ vs. $\theta$). All models were trained on $M = 15$ training points. Energies and errors in kcal/mol. A *black cross* marks the PBE equilibrium position

angle $\theta$ between ±8.59° (±0.15 rad) away from $\theta_0$ (see Supplementary Fig. 1 for a visualization of the sampled range). For this molecule, the out-of-sample test set again comprises a random subset of 50 geometries, with the remaining 300 geometries used as the grand training set. Because there are now three parameters, it is more difficult to select equidistant samples for the training subset of $M$ data points. We therefore use a K-means approach to find $M$ clusters and select the grand training set geometry closest to each cluster's center for the training subset (see Supplementary Note 4 for details).

Models are trained as for H₂. The results are given in Table 2. As expected, the increase in degrees of freedom for H₂O compared to H₂ requires a larger training set size $M$. However, even for the more complicated molecule, the ML-HK map is consistently more precise than the ML-KS map and provides an

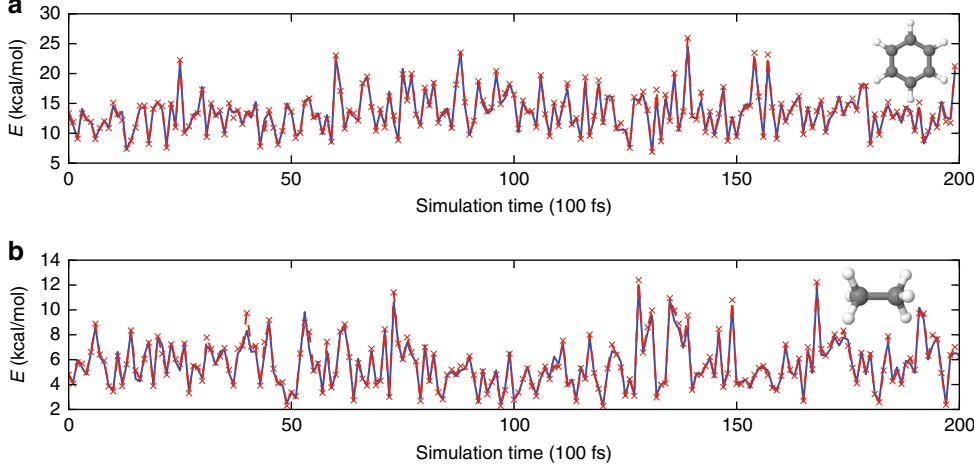

**Fig. 3** Energy errors of ML-HK along classical MD trajectories. PBE values in *blue*, ML-HK values in *red*. **a** A 20 ps classical trajectory of benzene. **b** A 20 ps classical trajectory of ethane

improved potential energy surface, as shown in Fig. 2. With an MAE of 1.2 kcal/mol for PBE energies relative to CCSD(T) calculations for this data set, we again show that ML does not introduce a new significant source of error.

The ML maps can also be used to find the minimum energy configuration. The total energy is minimized as the geometry varies with respect to both bond lengths and angles. For optimization, we use Powell's method[40], which requires a starting point and an evaluation function to be minimized. For the $H_2O$ case, the search is restricted to symmetric configurations with a random symmetric geometry used as the starting point. Results are reported in Table 2. The optimizations consistently converge to the correct minimum regardless of starting point, consistent with the maps being convex, i.e., the potential energy curves are sufficiently smooth to avoid introducing artificial local minima.

For larger molecules, generating random conformers that sample the full configurational space becomes difficult. Therefore, we next demonstrate that MD using a classical force field can also be used to create the grand training set. As an example, we use benzene ($C_6H_6$) with only small fluctuations in atomic positions out of the molecular plane (Supplementary Fig. 2). Appropriate conformers are generated via isothermal MD simulations at 300, 350, and 400 K using the General Amber Force Field (GAFF)[41] in the PINY_MD package[42]. Saving snapshots from the MD trajectories generates a large set of geometries that are sampled using the K-means approach to obtain 2000 representative points for the grand training set. Training of $n^{ML}[v]$ and $E^{ML}[n]$ is performed as above by running DFT calculations on $M = 2000$ points. We find that the ML error is reduced by creating the training set from trajectories at both the target temperature and a higher temperature to increase the representation of more distorted geometries. The final ML model is tested on 200 conformational snapshots taken from an independent MD trajectory at 300 K (Fig. 3a). The MAE of the ML-HK map for this data set using training geometries from 300 and 350 K trajectories is only 0.37 kcal/mol for an energy range that spans more than 10 kcal/mol (Table 3).

For benzene, we further quantify the precision of the ML-HK map in reproducing PBE densities. In Fig. 4, it is clear that the errors in the Fourier basis representation are larger than the errors introduced by the ML-HK map by two orders of magnitude. Furthermore, the ML-HK errors in density (as evaluated on a grid in the molecular plane of benzene) are also considerably smaller than the difference in density between density functionals (PBE vs. LDA[43]). This result verifies that the

**Table 3 Energy and density-driven errors of the ML-HK approach on the MD data sets**

| Molecule | Training trajectories | $\Delta E$ | | $\Delta E_D^{ML}$ | |
|---|---|---|---|---|---|
| | | MAE | Max | MAE | Max |
| Benzene | 300 K | 0.42 | 1.7 | 0.32 | 1.5 |
| | 300 + 350 K | 0.37 | 1.8 | 0.28 | 1.5 |
| | 300 + 400 K | 0.47 | 2.3 | 0.30 | 1.8 |
| Ethane | 300 K | 0.20 | 1.5 | 0.17 | 1.3 |
| | 300 + 350 K | 0.23 | 1.4 | 0.19 | 1.1 |
| | 300 + 400 K | 0.14 | 1.7 | 0.098 | 0.62 |
| Malonaldehyde | 300 + 350 K | 0.27 | 1.2 | 0.21 | 0.74 |

Errors are given in kcal/mol for different training trajectory combinations

ML-HK map is specific to the density used to train the model and should be able to differentiate between densities generated with other electronic structure approaches.

Ethane ($C_2H_6$), with a small energy barrier for the relative rotation of the methyl groups, is also evaluated in the same way. Using geometries sampled with K-means from 300 and 350 K classical trajectories, the ML-HK model reproduces the energy of conformers with a MAE of 0.23 kcal/mol for an independent MD trajectory at 300 K (Fig. 3b). This test set includes conformers from the sparsely-sampled eclipsed configuration (Supplementary Fig. 3). Using points from a 400 K trajectory improves the ML-HK map due to the increased probability of higher energy rotamers in the training set (Table 3). The training set could also be constructed by including explicit rotational conformers, as is commonly done when fitting classical force field parameters[41]. In either case, generating appropriate conformers for training via computationally cheap classical MD significantly decreases the cost of the ML-HK approach.

As additional proof of the versatility of the ML-HK map, we show that this approach is also able to interpolate energies for proton transfer in the enol form of malonaldehyde ($C_3H_4O_2$). This molecule is a well-known example of intramolecular proton transfer, and our previous AIMD and ab initio path integral studies[44] found classical and quantum-free energy barrier values of 3.5 and 1.6 kcal/mol, respectively, from gradient-corrected DFT. In this work, classical MD trajectories are run for each tautomer separately, with a fixed bonding scheme, then combined for K-means sampling to create the grand training set. The training set also includes an artificially constructed geometry that

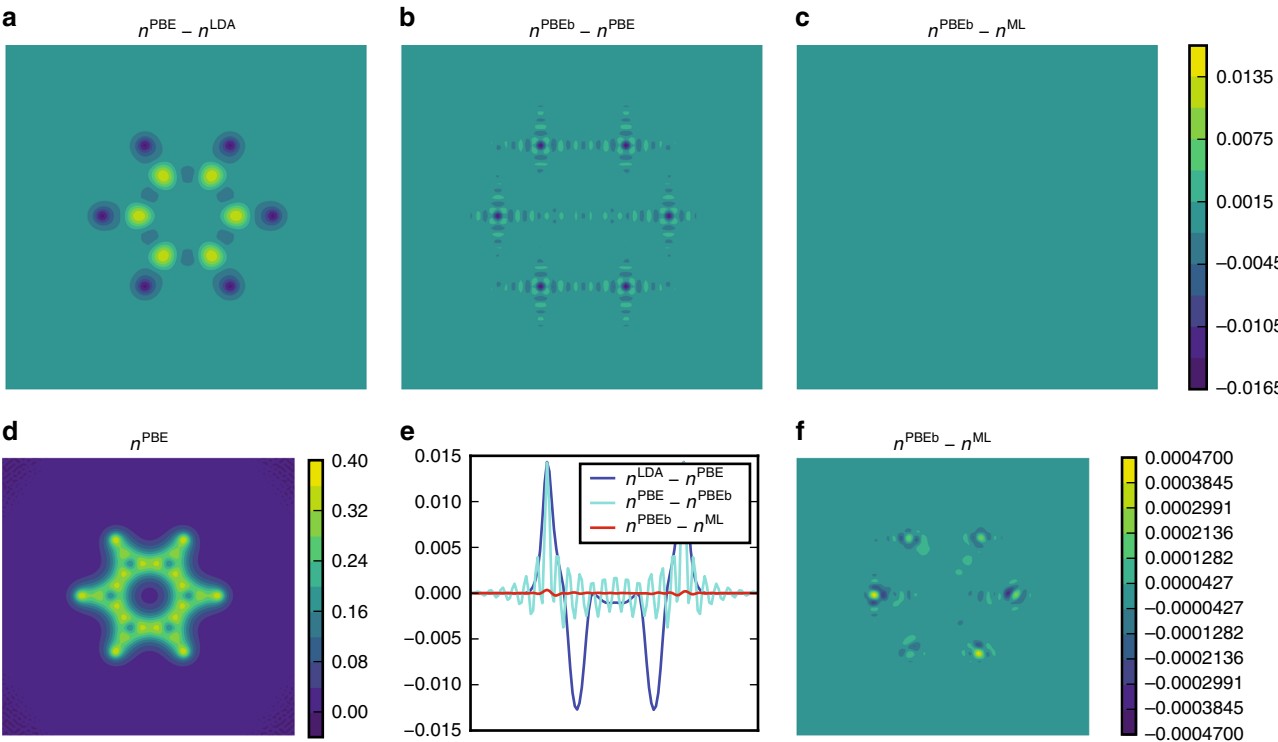

Fig. 4 The precision of our density predictions using the Fourier basis for the molecular plane of benzene. The plots show **a** the difference between the valence density of benzene when using PBE and LDA functionals at the PBE-optimized geometry. **b** Error introduced by using the Fourier basis representation. **c** Error introduced by the $n^{ML}[v]$ density fitting (**a**–**c** on same color scale). **d** The total PBE valence density. **e** The density differences along a 1D cut of **a**–**c**. **f** The density error introduced with the ML-HK map (same data but different scale, as in **c**)

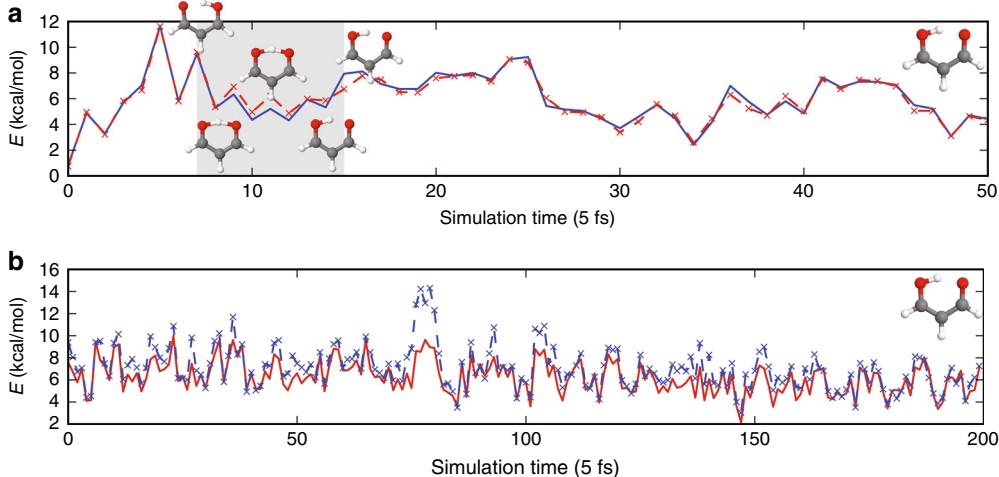

Fig. 5 Energy errors of ML-HK along ab initio MD and ML-generated trajectories. **a** Energy errors of ML-HK along a 0.25 ps ab initio MD trajectory of malonaldehyde. PBE values in *blue*, ML-HK values in *red*. The ML model correctly predicts energies during proton transfer in frames 7–15 without explicit inclusion of these geometries in the training set. **b** Energy errors of ML-HK along a 1 ps MD trajectory of malonaldehyde generated by the ML-HK model. ML-HK values in *red*, PBE values of trajectory snapshots in *blue*

is the average of tautomer atomic positions. For the test set, we use snapshots from a computationally expensive Born–Oppenheimer ab initio MD trajectory at 300 K. Figure 5a shows that the ML-HK map is able to predict DFT energies during a proton transfer event (MAE of 0.27 kcal/mol) despite being trained on classical geometries that did not include these intermediate points.

We show, finally, that the ML-HK map can also be used to generate a stable MD trajectory for malonaldehyde at 300 K (Fig. 5b). In principle, analytic gradients could be obtained for each timestep, but for this first proof-of-concept trajectory, a finite-difference approach was used to determine atomic forces. The ML-HK-generated trajectory samples the same molecular configurations as the ab inito MD simulation (see Fig. 6 and Supplementary Table 1) with a mean absolute energy error of 0.77 kcal/mol, but it typically underestimates the energy for out-of-plane molecular fluctuations at the extremes of the classical training set (maximum error of 5.7 kcal/mol, see Supplementary Fig. 4). Even with the underestimated energy values, however, the atomic forces are sufficiently large to return the molecule to the

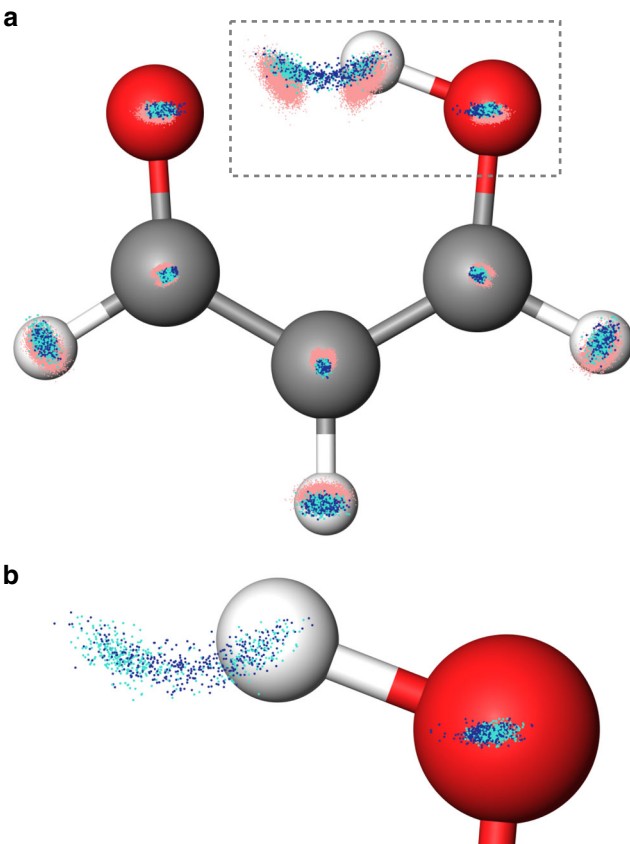

**Fig. 6** The space of malonaldehyde conformers generated by all MD methods. **a** The training set of 2000 representative conformers selected from the classical MD trajectories (*red points*) by K-means sampling. Test points from an ab initio MD trajectory (*turquoise*) and the independently generated MD trajectory using the ML-HK model (*blue*) sample the same coordinate space (offset from the molecular plane for clarity). **b** A closer view of the region outlined with a *dashed box* for the ab initio (*turquoise*) and ML-HK (*blue*) trajectories

equilibrium configuration, thus resulting in a stable trajectory. The new set of coordinates could be further sampled to expand the training set in a self-consistent manner. Using iterative ML-HK-generated MD trajectories would eliminate the need to run computationally expensive MD simulations with KS DFT and would provide an iterative approach to reduce the energy errors for conformations not included in the classical training set.

## Discussion

For several decades, DFT has been a cross-disciplinary area between theoretical physics, chemistry, and materials sciences. The methods of each field cross-fertilize advances in other fields. This has led to its enormous popularity and widespread success, despite its well-known limitations in both accuracy and the systems and properties to which it can be applied.

The present work represents a key step forward toward adding an additional ingredient to this mix, namely the construction of functionals via ML. While previous work showed proofs of principle in 1D, this work is a demonstration in 3D, using real molecules and production-level codes. We also demonstrate that molecular conformers used in the training set can be generated by a range of methods, including informed scans and classical MD simulations. This opens the possibility that ML methods, which complement all existing approaches to functional

approximation, could become a new and very different approach to this problem with the potential to reduce the computational cost of routine DFT calculations significantly.

Our method, directly learning the HK density-potential map, overcomes a major bottleneck in previous methodologies that arises in 3D. Our approach avoids solving an intermediate more general problem (the gradient descent) to find the solution of the more specific problem (finding the ground-state density). This is called transductive inference by the ML community and is thought to be key to the success of statistical inference methods[45]. Following a direct prediction approach with the ML-HK map increases the accuracy consistently on both 1D examples and 3D molecules. We are also able to learn density models that out-perform energy models trained on much more data. This quantitative observation allows us to conclude that learning density models is much easier than learning energy models. Such a finding should be no surprise to practitioners of the art of functional construction (see, e.g., ref. [28]), but the present work quantifies this observation using standard statistical methods. As the ML-HK map accurately reflects the training densities, more exact methods could also be used to generate the training set densities for functional development.

We have also derived a scheme for using basis functions to render the approach computationally feasible, which allows for facile integration of the method into existing DFT codes. Another advantage is the possibility to take the innate structure of the densities into account, i.e., spatial correlations are preserved by using low-frequency basis functions. Again, this fits with the intuition of experienced practitioners in this field, but here we have quantified this in terms of machine-learned functionals.

Direct prediction of energies (e.g., the ML-KS map) always has the potential to lead to conceptually easier methods. But such methods must also abandon the insights and effects that have made DFT a practical and usefully accurate tool over the past half century. Many usefully accurate DFT approximations already exist, and the corrections to such approximations can be machine-learned in precisely the same way as the entire functional has been approximated here[23]. If ML corrections require less data, the method becomes more powerful by taking advantage of existing successes. Furthermore, existing theorems, such as the viral theorem[46], might also be used to construct the kinetic energy functional directly from an ML-HK map. In the case of orbital-dependent functionals, such as meta-GGA's or global hybrids, the method presented here must be extended to learn, e.g., the full density matrix instead of just the density.

We also note that, for all the 3D calculations shown here, we machine-learned $E[n]$, the entire energy (not just the kinetic energy), which includes some density-functional approximation for XC. However, with a quantum chemical code, we could have trained on much more accurate quantum chemical densities and energies. Thus, the ML-HK maps, in principle, allow the construction of (nearly) exact density functionals for molecular systems, with the potential to significantly reduce the computational cost of quantum chemistry-based MD simulations. All this points to useful directions in which to expand on the results shown here.

## Methods

**Kohn–Sham DFT**. The KS-DFT computational electronic structure method determines the properties of many-body systems using functionals of the electron density. The foundation is the HK theorem[47], which establishes a one-to-one relationship between potential and density, i.e., at most one potential can give rise to a given ground-state density.

Kohn–Sham DFT avoids direct approximation of many-body effects by imagining a fictitious system of non-interacting electrons with the same density as the real one[1] (see Supplementary Note 5 for details). Its accuracy is limited by the accuracy of existing approximations to the unknown XC energy, while its

computational bottleneck is the solution of the Kohn–Sham equations that describe the non-interacting particles.

Here, 3D DFT calculations for ML models are performed with the Quantum ESPRESSO code[25] using the PBE XC functional[36] and PAWs[37, 38] with Troullier–Martins pseudization for describing the ionic cores[48]. All molecules are simulated in a cubic box ($L = 20$ bohr) with a wave function cutoff of 90 Ry (see Supplementary Note 3 for details). The 1D data set is taken from ref. [20].

**Kernel ridge regression**. KRR[49, 50] is a ML method for regression. It is a kernelized version of ridge regression that minimizes the least squares error and applies an $\ell_2$ (Tikhonov) regularization. Let $\mathbf{x}_1, \dots, \mathbf{x}_m \in \mathbb{R}^d$ be the training data points and let $\mathbf{Y} = (\mathbf{y}_1, \dots, \mathbf{y}_m)^{\mathbf{T}}$ be their respective labels. KRR then performs the following optimization:

$$\min_\alpha \sum_{i=1}^m \left| \mathbf{y}_i - \sum_{j=1}^m \boldsymbol{\alpha}_j k(\mathbf{x}_i, \mathbf{x}_j) \right|^2 + \lambda \boldsymbol{\alpha}^{\mathbf{T}} \mathbf{K} \boldsymbol{\alpha}, \qquad (13)$$

where $k$ is the kernel function and $\lambda$ is a regularization parameter. $\mathbf{K}$ is the kernel matrix with $\mathbf{K}_{ij} = k(\mathbf{x}_i, \mathbf{x}_j)$. Eq. (13) has the following analytical solution:

$$\boldsymbol{\alpha} = (\mathbf{K} + \lambda \mathbf{I})^{-1} \mathbf{Y}. \qquad (14)$$

Most popular is the Gaussian (radial basis function) kernel that leads to a smooth, non-linear model function in input space corresponding to a linear function in an infinite dimensional feature space[29].

For the ML-HK map (Supplementary Methods), the canonical error is given by the $\mathcal{L}_2$ distance between predicted and true densities

$$e(\boldsymbol{\beta}) = \sum_{i=1}^M \left\| n_i - n^{\mathrm{ML}}[v_i] \right\|_{\mathcal{L}_2} \qquad (15)$$

$$= \sum_{i=1}^M \left\| n_i - \sum_{l=1}^L \sum_{j=1}^M \beta_j^{(l)} k(v_i, v_j) \phi_l \right\|_{\mathcal{L}_2}. \qquad (16)$$

The ML model coefficients $\boldsymbol{\beta}^{(l)}$ can be optimized independently for each basis coefficient $l$ via

$$\boldsymbol{\beta}^{(l)} = \left( \mathbf{K}_{\sigma^{(l)}} + \lambda^{(l)} \mathbf{I} \right)^{-1} \mathbf{u}^{(l)}, \quad l = 1, \dots, L. \qquad (17)$$

**Cross-validation**. Note that all model parameters and hyper-parameters are estimated on the training set; the hyper-parameter choice makes use of standard cross-validation procedures (see Hansen et al.[12]). Once the model is fixed after training, it is applied unchanged out of sample.

**Exact calculations**. Relative energy errors of the ML models trained on KS-DFT calculations are determined by comparing to accurate energies from the Molpro Quantum Chemistry Software[51] using the full configuration interaction method for $H_2$ and CCSD(T)[52] for $H_2O$.

**Molecular dynamics**. For benzene, ethane, and malonaldehyde, GAFF parameters[41] were assigned using the AmberTools package[53]. Geometry optimizations were performed using MP2/6-31g(d) in Gaussian09[54]. Atomic charge assignments are from RESP fits[55] to HF/6-31g(d) calculations at optimized geometries and two additional rotational conformers for ethane.

For the three larger molecules, classical isothermal MD simulations were run using the PINY_MD package[42] with massive Nosé–Hoover chain (NHC) thermostats[56] for atomic degrees of freedom (length = 4, $\tau$ = 20 fs, Suzuki–Yoshida order = 7, multiple time step = 4) and a time step of 1 fs. The r-RESPA multiple time step approach[57] was employed to compute rapidly varying forces more frequently (torsions every 0.5 fs, bonds/bends every 0.1 fs). Systems were equilibrated for 100 ps before collecting snapshots every 100 fs from 1 ns trajectories. Snapshots were aligned to a reference molecule prior to DFT calculations for the ML model. For malonaldehyde, the ML training set geometries were selected from trajectories for both enol tautomers as the GAFF force field does not permit changes in chemical bond environments.

For malonaldehyde, an additional Born–Oppenheimer MD simulation using DFT was run using the QUICKSTEP package[58] in CP2K v. 2.6.2[59]. The PBE XC functional[36] was used in the Gaussian and plane-wave scheme[60] with DZVP-MOLOPT-GTH (m-DZVP) basis sets[61] paired with the appropriate dual-space GTH pseudopotentials[62] optimized for the PBE functional[63]. Wave functions were converged to 1E-7 Hartree in the energy using the orbital transformation method[64] on a multiple grid ($n = 5$) with a cutoff of 900 Ry for the system in a cubic box ($L = 20$ bohr). A temperature of 300 K was maintained using massive NHC thermostats[56] (length = 4, $\tau$ = 10 fs, Suzuki–Yoshida order = 7, multiple time step = 4) and a time step of 0.5 fs.

In order to generate the MD trajectory with the ML-HK model, we used the Atomistic Simulation Environment[65] with a 0.5 fs timestep and a temperature of 300 K maintained via a Langevin thermostat. A thermostat friction value of 0.01 atomic units (0.413 fs$^{-1}$) was chosen to reproduce the fluctuations in C atoms observed for the DFT-based trajectory (Supplementary Note 3). In this proof-of-concept work, atomic forces were calculated using central finite differences, with $\epsilon = 0.001$ Å chosen to conserve the total energy during the simulation. The last 1 ps of a 4 ps trajectory was used to evaluate the performance of the ML-HK model.

**Data availability**. All data sets used in this work are available at http://quantum-machine.org/datasets/.

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

## Acknowledgements

We thank J.C. Snyder for early discussions and H. Glawe for helpful guidance regarding the 3D reference computations. We thank IPAM at UCLA for repeated hospitality. Work at UC Irvine supported by NSF CHE-1464795. K.R.M. and F.B. thank the Einstein Foundation for generously funding the ETERNAL project. This work was supported by Institute for Information & Communications Technology Promotion (IITP) grant funded by the Korea government (no. 2017-0-00451). Work at NYU supported by the US Army Research Office under contract/grant number W911NF-13-1-0387 (M.E.T. and L.V.). Ab initio trajectories were run using High Performance Computing resources at NYU. Other DFT simulations were run using High Performance Computing resources at MPI Halle.

## Author contributions

F.B. performed DFT calculations and ML experiments. L.V. performed classical and ab initio MD simulations. L.L. performed FCI and CCSD(T) calculations. K.B. and K-R.M. initiated the work, and K.B., K-R.M., and M.E.T. contributed to the theory and experiments. All authors contributed to the manuscript.

## Additional information

**Competing interests:** The authors declare no competing financial interests.

