## [Peer Review File · Nature Communications]

Reviewers' comments:

Reviewer #1 (Remarks to the Author):

The manuscript explores a way to use machine learning (ML) to aid density functional theory (DFT) calculations. DFT is based on the theorem that the ground state energy of an interacting system of electrons in an external potential is a functional of just the total electron density, and the density corresponding to the ground state is in one to one correspondence with the external potential. However, important parts of this functional are not known (specifically, the correlation energy, kinetic energy). Nevertheless, good approximations for this functional have seen DFT gain widespread use. Given such an approximate functional, the energy is minimised over the allowable densities (or sometimes corresponding one-electron wavefunctions), a procedure that is still quite costly. A longstanding goal of some of the authors has been to use machine learning to bypass some of these calculations. In earlier work, they attempted to approximate the kinetic energy expression based on wave functions by one that is based on densities using a machine learning framework. In the present manuscript, they use machine learning to approximate the ground state density as a functional of the external potential. If successful, it would represent a key step in a larger programme of speeding up electronic structure calculations.

In my view, the manuscript represents a modest trialling of its central idea. Its results are mildly encouraging, and will no doubt prompt further research by its authors to expand on the present results and possibly interest some others in the small community of researchers who are playing with machine learning ideas in the context of materials simulation.

Some specific criticisms, in no particular order:

1) Even if accurate densities were obtainable from external potentials in general using this method, there remains the problem of evaluating the energy given a density. The authors subtly acknowledge this on line 246, and mention in passing that they trained an energy model on densities to be able to benchmark their potential->density map. Scant details are given about this critical piece.

2) The methods are tried on very small systems and very small training and test sets. Experience has shown that ML models are so powerful and flexible, that it is very easy to show good fits on very small datasets, that do not explore a wide range of input data. The presented results would not convince any practitioners that the method WILL work on a larger scale. It MIGHT work, and thus the manuscript represents the first successful foray into using this approach, but much more is required to call it a useful method.

3) The method as presented has no transferability from one system to another. The map trained for a water molecule will only work for an isolated water molecule. This means that the elusive goal of training the potential->density map (or the density->energy map) is no closer, for each new system data will need to be generated. How much data? how does the required data scale with system size? No-one knows. Answers to these questions will make or break the method in practice. Only future work will tell.

4) The manuscript is rather poorly written, and is difficult to follow. E.g. no details are given for the ML-KS method that is being used as reference. Equations are presented without explanation of their symbols. CI and CCSD(T) methods are mentioned in the methods section, but I found no use of them in the manuscript.

In summary, the manuscript is of moderate interest to a small and specialised community, and falls far, far short of impact and importance to be published in Nature Comms.

Reviewer #2 (Remarks to the Author):

This paper proposes a method for efficiently predicting the DFT total energy from a given external potential, based on a machine learning (ML) potential-density map. This method is applied to 1-D model potential system and 3-D molecules modeled by artificial Gaussian potentials. Their method enables accelerating the geometry optimization and ab initio molecular dynamics significantly, which have been performed by so many researchers.

The authors' idea that the relationship between total energy and external potential is estimated via a direct potential-electron density map is very interesting. And it is a surprising result that the electron density can be well described by kernel ridge model made from the external potential. In addition, this paper is well-written and well-organized.

On the other hand, this method can be regarded as a method for predicting potential energy surfaces (PES) or interatomic potentials in the referee's understanding. Many studies on ML-PES or interatomic potentials in molecules and solids have been published so far. (For example, molecules: JCP 85, 911 (1986), JCP 126, 184108 (2007), CPL 395, 210 (2004). solids: PRL 98, 146401 (2007), PRL 104, 136403 (2010), PRB 90, 024101 (2014)) Although the quality of this paper is no doubt high, the referee thinks that obtained results are too primitive, compared to those previous studies. So the referee suggests to include more practical applications to get an interest of a broad readership.

Minor comments and questions are as follows.

Descriptions on how to approximate $E[n]$ and how to perform direct Kohn-Sham mapping are lacking, although the former is described in the supplements. The referee recommends to include a brief description on them in the main text.

How to use the external potential in the Gaussian kernel? Is the external potential in a grid form used, and then is the difference of multi-dimensional vectors for the two external potentials considered in the Gaussian kernel?

Dear reviewers,

First, we would like to summarize all of the major revisions that have been included in the new ms. Then we address each reviewer's comments point-by-point.

Energy map $E^{ML}[\mathbf{n}]: \mathbf{n} \rightarrow E$

Our earlier work showed that it is relatively easy to learn the energy highly accurately, once the density is known. We now verify this also for 3-D molecules by training all energy maps on the same training points used for the ML-HK map. We completely removed all energy maps that were trained on more training points.

We further updated all 3-D figures to display the combined error of the ML-HK map and energy map.

Density-driven error ΔE_D and ΔE_D^{ML}

We updated the method to estimate the density-driven error of our ML-HK map. We previously used energy maps trained on significantly more training points (as a proxy to ground truth) to estimate the density-driven error. Answering to the reviewers concerns, we removed all these energy maps and

- use the von Weizsäcker kinetic energy to compute the exact density-driven error (ΔE_D) in 1-D and
- use energy maps trained on the same training points as the ML-HK map to estimate the density-driven error (ΔE_D^{ML}) in 3-D.

Experiments

We included experiments on bigger and chemically more complex molecules: Benzene, Ethane, and Malonaldehyde.

Sampling strategy

We introduced a method to sample the configurational space in order to reduce the amount of DFT calculations that are necessary as training points for machine learning. We use

molecular dynamics simulations with classical force fields and show that higher temperature simulations aid in representing more distorted geometries.

ML-KS map, external potential, and the Gaussian kernel

We clarified the ML-KS model and how the external potential $v(r)$ is applied in the Gaussian kernel.

Density predictions

We added results comparing the errors made by our ML-HK map to differences between LDA and PBE densities. We also extended the discussion section to emphasize that the methodology could equally be applied to training data generated by more accurate quantum chemical codes.

(Point-by-point comments within.)

Reviewer #1 (Remarks to the Author):

The manuscript explores a way to use machine learning (ML) to aid density functional theory (DFT) calculations. DFT is based on the theorem that the ground state energy of an interacting system of electrons in an external potential is a functional of just the total electron density, and the density corresponding to the ground state is in one to one correspondence with the external potential. However, important parts of this functional are not known (specifically, the correlation energy, kinetic energy). Nevertheless, good approximations for this functional have seen DFT gain widespread use. Given such an approximate functional, the energy is minimised over the allowable densities (or sometimes corresponding one-electron wavefunctions), a procedure that is still quite costly. A longstanding goal of some of the authors has been to use machine learning to bypass some of these calculations. In earlier work, they attempted to approximate the kinetic energy expression based on wave functions by one that is based on densities using a machine learning framework. In the present manuscript, they use machine learning to approximate the ground state density as a

functional of the external potential. If successful, it would represent a key step in a larger programme of speeding up electronic structure calculations.

In my view, the manuscript represents a modest trialling of its central idea. Its results are mildly encouraging, and will no doubt prompt further research by its authors to expand on the present results and possibly interest some others in the small community of researchers who are playing with machine learning ideas in the context of materials simulation.

First of all, we would like to thank for the excellent feedback that we received for our ms. It allowed us to improve the presentation of our material on one side but perhaps even more importantly, we have added novel results that substantially improve and extend the ms, hopefully convincing the reviewer of its broader applicability that we clearly see of general interest.

Some specific criticisms, in no particular order:

1) Even if accurate densities were obtainable from external potentials in general using this method, there remains the problem of evaluating the energy given a density. The authors subtly acknowledge this on line 246, and mention in passing that they trained an energy model on densities to be able to benchmark their potential->density map. Scant details are given about this critical piece.

Our earlier work showed that it is relatively easy to learn the energy highly accurately, once the density is known. This is also verified by table 1. It requires less data than for learning the density. This is why we did not further emphasize this aspect in the current manuscript, however we have now added a paragraph clarifying this point raised and hope that we could make this substantially clearer.

In the revised ms, for the toy model, we now show that, when the same number of data points is used for both the energy and density maps, the functional- and density-driven errors are comparable, so that one does not need a very highly-trained energy functional for the energy evaluation. In the revised ms, for all molecular calculations, we now use the

same number of training points for the energy and density evaluations, and get accuracies very similar to (but slightly worse than) previously. So this problem has been circumvented. We also note that we discovered our previous evaluation of the density-driven error for the toy model was a substantial overestimate, so that in fact our density-driven errors appear to be significantly smaller.

2) The methods are tried on very small systems and very small training and test sets. Experience has shown that ML models are so powerful and flexible, that it is very easy to show good fits on very small datasets, that do not explore a wide range of input data. The presented results would not convince any practitioners that the method WILL work on a larger scale. It MIGHT work, and thus the manuscript represents the first successful foray into using this approach, but much more is required to call it a useful method.

We entirely agree that the current implementation is limited to small molecules, as is stated in the manuscript. But our new material on MD simulations shows that, even with this limitation, useful science can be done, just not yet on bulk materials. So the reviewer may forgive us that we strongly disagree with the statement that the method is not useful in its current form.

3) The method as presented has no transferability from one system to another. The map trained for a water molecule will only work for an isolated water molecule. This means that the elusive goal of training the potential->density map (or the density->energy map) is no closer, for each new system data will need to be generated. How much data? how does the required data scale with system size? No-one knows. Answers to these questions will make or break the method in practice. Only future work will tell.

While it is true that the ML functional does not transfer to the full chemical compound space, this is not a limitation of this method in the way we envision it being used. We imagine that, for each new system, one trains a new map, and then applies it to the system under study, and even discards it at the end. The revised manuscript now illustrates this process for several small molecules. Of course, training costs are always an important determinant of whether or not the method will be useful for a particular situation. Generally

speaking, ML training costs are rather negligible. Clearly we have so far not entered the realm of very large systems, so we cannot provide quantitative results whether training costs may eventually become an issue. Note however, that the novel procedure introduced which uses classical trajectories and informed sampling to cover the configurational space in an intelligent manner already shows a first practical path for larger systems, otherwise the novel studies including Benzene, Ethane, as well as more complex/chemically interesting systems Malonaldehyde could not have been done.

4) The manuscript is rather poorly written, and is difficult to follow. E.g. no details are given for the ML-KS method that is being used as reference. Equations are presented without explanation of their symbols. CI and CCSD(T) methods are mentioned in the methods section, but I found no use of them in the manuscript.

It is true that we do not give details for the ML-KS method in the main text, because that appears in earlier publications, and space is limited. All symbols are defined in this or previous manuscripts, or are in common use. We have revised and extended the manuscript substantially. We added Eq. 6 to clarify that the ML-KS map is a kernel model and significantly extended the section on motivation and explanation of Kernel Ridge Regression in the supplement. The CI and CCSD(T) calculations were only used to make the reference curves for H₂ and the benchmark values for H₂O, as is now made clear.

In summary, the manuscript is of moderate interest to a small and specialised community, and falls far, far short of impact and importance to be published in Nature Comms.

We strongly hope that our improved presentation and the new material on MD simulations added will convince the skeptical reviewer that useful science (even more interesting chemistry, see Malonaldehyde) can be indeed be done, and that there is a clearer path to broader applications.

Reviewer #2 (Remarks to the Author):

This paper proposes a method for efficiently predicting the DFT total energy from a given external potential, based on a machine learning (ML) potential-density map. This method is

applied to 1-D model potential system and 3-D molecules modeled by artificial Gaussian potentials. Their method enables accelerating the geometry optimization and ab initio molecular dynamics significantly, which have been performed by so many researchers.

The authors' idea that the relationship between total energy and external potential is estimated via a direct potential-electron density map is very interesting. And it is a surprising result that the electron density can be well described by kernel ridge model made from the external potential. In addition, this paper is well-written and well-organized.

First of all, we would like to thank for the excellent feedback that we received for our ms. It allowed us to improve the presentation of our material on one side but perhaps even more importantly, we have added novel results that substantially improve and extend the ms.

On the other hand, this method can be regarded as a method for predicting potential energy surfaces (PES) or interatomic potentials in the referee's understanding. Many studies on ML-PES or interatomic potentials in molecules and solids have been published so far. (For example, molecules: JCP 85, 911 (1986), JCP 126, 184108 (2007), CPL 395, 210 (2004). solids: PRL 98, 146401 (2007), PRL 104, 136403 (2010), PRB 90, 024101 (2014)) Although the quality of this paper is no doubt high, the referee thinks that obtained results are too primitive, compared to those previous studies. So the referee suggests to include more practical applications to get an interest of a broad readership.

We include a selection of the references which makes the ms have a broader appeal. In addition, answering to the concern of practical applicability, we added substantial novel material on larger and more complex systems, i.e. Benzene, Ethane, Malonaldehyde. We would however also like to clarify that what we propose should not be conceived primarily as a method for finding energies and forces. We see it instead as a method aimed at providing a density functional approximation that can be applied to large systems with the accuracy of present-day DFT calculations (and more accurate methods). While this step has not yet been achieved, the current work shows necessary (and we modestly think substantial) progress in that direction, and overcomes difficulties of previous methods in finding the functional derivative. Once the method is generalized to larger systems, such as

bulk water, via the density, then it can be applied to as diverse array of systems as are tackled with present KS-DFT calculations, but at a fraction of the cost, allowing much larger time scales to be explored, whereas most force field methods are not general and need to be rebuilt for different elements or phases.

We also now emphasize that our methodology could equally be applied to training data generated by quantum chemical methods, thereby generating density functionals of much higher accuracy than standard DFT approximations. We are already testing this idea in further work.

Minor comments and questions are as follows.

Descriptions on how to approximate $E[n]$ and how to perform direct Kohn-Sham mapping are lacking, although the former is described in the supplements. The referee recommends to include a brief description on them in the main text.

We extended the main text of our ms by adding Eq. 6 and a paragraph that explains this in more detail.

How to use the external potential in the Gaussian kernel? Is the external potential in a grid form used, and then is the difference of multi-dimensional vectors for the two external potentials considered in the Gaussian kernel?

We indeed use the external potential exactly as the reviewer suspected, but have, in the interest of removing any ambiguity, clarified this section in the manuscript.

Reviewers' comments:

Reviewer #1 (Remarks to the Author):

The authors have significantly extended the numerical experiments in their manuscript, and this is very commendable. The system sizes involved now offer a suggestion that their method is indeed useful for realistic systems of interest, and might be able to significantly speed up ab initio molecular dynamics simulations. In the authors' conception, the strategy would be to run molecular dynamics with a cheap model (or a short trajectory with DFT), collect density and energy data, fit two models (potential->density and density->energy), and then use the ML model to run a much longer molecular dynamics trajectory which has ab initio quality but much much cheaper.

All the required numerical tools have been assembled and demonstrated in the manuscript, so my question is: why have the authors actually not done this? What happens when the model of malonaldehyde is used to generate the trajectory? Is it stable? is the ML model accurate when measured on samples generated by itself? (Fig 3 only shows their model is accurate when evaluated on samples from a classical model trajectory.) If the MD trajectory based on the ML model is good (i.e. that the ML predictions are good on samples from this trajectory), I'm satisfied that the manuscript represents a significant and practical advance. If not, then more work is needed to identify what the problem is.

It is conceivable that the gradients of the ML model with respect to atomic positions have not yet been implemented in code, and which are necessary to carry out MD. This should not pose a practical problem, since a simple finite difference approximation of the gradient can be effectively used for the purpose of this test, since the number of atoms is small, and the model is fast.

Reviewer #2 (Remarks to the Author):

The authors have satisfactorily addressed my major comments as well as minor ones. They show more practical examples in the revised manuscript. Therefore, I think the manuscript is suitable for publication on Nature Communications.

Dear reviewers,

Thank you for reviewing our revised manuscript "By-passing the Kohn-Sham equations with machine learning". Please refer to point-by-point comments within.

Best regards,

The Authors

Reviewer #1 (Remarks to the Author):

The authors have significantly extended the numerical experiments in their manuscript, and this is very commendable. The system sizes involved now offer a suggestion that their method is indeed useful for realistic systems of interest, and might be able to significantly speed up ab initio molecular dynamics simulations. In the authors' conception, the strategy would be to run molecular dynamics with a cheap model (or a short trajectory with DFT), collect density and energy data, fit two models (potential->density and density->energy), and then use the ML model to run a much longer molecular dynamics trajectory which has ab initio quality but much much cheaper.

All the required numerical tools have been assembled and demonstrated in the manuscript, so my question is: why have the authors actually not done this? What happens when the model of malonaldehyde is used to generate the trajectory? Is it stable? is the ML model accurate when measured on samples generated by itself? (Fig 3 only shows their model is accurate when evaluated on samples from a classical model trajectory.) If the MD trajectory based on the ML model is good (i.e. that the ML predictions are good on samples from this trajectory), I'm satisfied that the manuscript represents a significant and practical advance. If not, then more work is needed to identify what the problem is.

First, we would like to point out that the revised manuscript already included an evaluation of the ML-HK map performance on an ab initio trajectory of malonaldehyde in Fig. 3c. To clarify the presentation, we separated the trajectories generated via classical force fields from the trajectories generated via ab initio molecular dynamics and the previous Fig. 3c is now presented as Fig. 5a.

In addition, we followed the reviewer's suggestion and generated, as a proof of concept, a molecular dynamics trajectory using our ML-HK map (Fig. 5b). The trajectory is stable and accurate when measured on samples generated by itself, but typically underestimates the energy for out-of-plane molecular fluctuations at the extremes of the classical training set. Revised and/or adaptive sampling procedures could further improve the prediction results, but since the sampling procedure itself is not the focus of our current manuscript, we believe this aspect merits an independent treatment in future work.

It is conceivable that the gradients of the ML model with respect to atomic positions have not yet been implemented in code, and which are necessary to carry out MD. This should not pose a practical problem, since a simple finite difference approximation of the gradient can be effectively used for the purpose of this test, since the number of atoms is small, and the model is fast.

The gradients have been implemented as suggested, via central finite differences (see manuscript for details).

Reviewer #2 (Remarks to the Author):

The authors have satisfactorily addressed my major comments as well as minor ones. They show more practical examples in the revised manuscript. Therefore, I think the manuscript is suitable for publication on Nature Communications.

The authors are pleased to have addressed all concerns and thank the reviewer for the helpful comments.

REVIEWERS' COMMENTS:

Reviewer #1 (Remarks to the Author):

The authors have done significant extra work to address my comment, I am happy for the manuscript to be published.

Dear reviewers,

Thank you for reviewing our manuscript “By-passing the Kohn-Sham equations with machine learning”.

Best regards,

The Authors

Reviewer #1 (Remarks to the Author):

The authors have done significant extra work to address my comment, I am happy for the manuscript to be published.

We thank the reviewer for his helpful comments that sparked significant improvements to our manuscript.